# Levels of Whole-Body Vibrations Transmitted to the Driver of a Tractor Equipped with Self-Levelling Cab during Soil Primary Tillage

Daniele Pochi, Laura Fornaciari, Gennaro Vassalini, Renato Grilli and Roberto Fanigliulo *

Consiglio per la Ricerca in Agricoltura e l'Analisi dell'Economia Agraria (CREA), Centro di Ricerca Ingegneria e Trasformazioni Agroalimentari (Research Centre for Engineering and Agro-Food Processing), Via della Pascolare 16, Monterotondo, 00015 Rome, Italy; daniele.pochi@crea.gov.it (D.P.); laura.fornaciari@crea.gov.it (L.F.); gennaro.vassalini@crea.gov.it (G.V.); renato.grilli@crea.gov.it (R.G.)
* Correspondence: roberto.fanigliulo@crea.gov.it; Tel.: +39-06-90675232

**Abstract:** Agricultural tractor drivers' health preservation and comfort represent important aspects of the evolution of agricultural machinery and led to the development of devices aimed at improving working conditions, such as soundproof cab and driver seat suspension, nowadays commonly adopted in tractors. The vibrations are one of the factors mostly affecting health and comfort conditions, resulting from the characteristics and interaction of specific tractor's parts (tyres, axles, chassis, cab). Trying to improve their products, manufacturers developed a cab prototype equipped with an automatic self-levelling system, whose goal is to maintain the driver's vertebral column in a correct position during heavy agricultural operations such as primary soil tillage. A tractor with a such a prototype was tested to assess its effectiveness in maintaining the cab horizontal and any effects on the transmitted levels of whole-body vibration, during soil primary tillage carried out by means of a mouldboard plough and a subsoiling plough, both in plain and hilly surfaces. The results showed that the device worked well at a slope lower than the operating limits of the system, keeping the cabin horizontal through progressive adjustments. A slight reduction of the level of vibration was observed with a self-levelling system working during the tillage tests in the plain, compared to the traditional condition.

**Keywords:** health preservation; whole-body vibrations; daily exposure time; mouldboard plough; subsoiling plough

## 1. Introduction

In recent years, beside the operative performances, the attention of the manufacturers of agricultural machinery has been increasingly focused on the aspects of comfort and health preservation of the operator, leading to the introduction of devices and instruments capable of significantly improving the working conditions. For example, modern agricultural tractor cabs are normally equipped with air conditioning, soundproofing systems [1] and driver seats with suspensions effective against the vibrations typical of agricultural work, responsible for temporary discomfort conditions. The vibrations transmitted to the driver's whole-body (WBV) are one of the factors most affecting health and comfort conditions. They are the result of the characteristics of elements such as cabin, tyres, chassis [2–4], axles [5,6], and seat suspensions [7–9] which differently interact depending on the external conditions (soil unevenness, slope, type of agricultural operation, speed, etc.).

In this regard, the manufacturers of tractors are trying to improve the health safeguard and comfort conditions in their products, through the development and introduction of new devices. Among them we find the cabin equipped with an automatic self-levelling system, the goal of which is to change the cockpit setting and maintain the driver in the correct position during agricultural operations that are more demanding on his spine. Driving

tractors, in fact, induces a postural overload, accentuated by any transverse slopes, due to the frequent rotations of the lumbar spine to perform operations in a sitting position [10].

Many researchers demonstrated that musculo-skeletal disorders (MSDs) in the driver workplace are caused by mechanical vibrations, and that there is a relationship between WBV exposure and MSDs, especially for low back pain (LBP), associated with increased risk of injury [11–14]. Few studies on WBV exposure in actual tractor-driving conditions are available [15,16]. Such risks become dangerous when the intensity of the vibrations is high, includes strong shocks or jolts, and occurs frequently with prolonged and repetitive exposure [17]. According to the Directive 2002/44/EC, tractor drivers' exposure to WBV should not exceed a daily action value of 0.5 m s$^{-2}$ (exposure action value, EAV) [18]. The cited Directive and the European Parliament established the minimum protection requirements for the workers exposed to the risks of vibration in the workplace.

When the WBV exceeds the EAV, actions to reduce the risks from vibration must be taken. On the other hand, the level of WBV is affected by the type of operation, the relating implement coupled to the tractor and the surface characteristics. For example, after harvesting, the presence of clods and deep cracks makes the soil remarkably hard and difficult to till and sow, particularly on slopes [19–21]. Primary tillage foresees the inversion of the soil layers, by means of reversible mouldboard plough, or a vertical shatter without mixing the tilled layers, carried out by implements such as a subsoiling plough, which aims to restore soil structure and to mitigate soil compaction [22,23]. The subsoiling is a principal substitutive of the ploughing and, differently from the ploughing, it favours the formation of the soil structure reducing superficial cloddiness [24], often making harrowing unnecessary.

This study concerned a series of tests on a medium-power tractor featuring a prototype of automated self-levelling cab with hydraulic control, which should allow the operator to maintain the correct vertical posture even during heavy tillage, such as deep ploughing in-furrow and subsoiling, on both plain and hilly surfaces, helping to reduce the effects of prolonged exposure to vibration on the spine in incorrect posture. The study was focused on the aspect of the driver's health to verify the effectiveness of the elastic systems in damping vibrations that are harmful to the human body. Therefore, the levels of vibration transmitted from the seat to the driver were measured according to the standard ISO 2631-1:1997 [25] in the frequency band 0.5–80 Hz considered more dangerous for the human body in a sitting position. The standard also specifies the location and direction of the measurements, the equipment to be used, the duration of the measurements and the frequency weighting, the measurement assessment methods and the evaluation of weighted root-mean-square (RSM) acceleration. The tests also aimed at determining the best efficiency conditions in maintaining the cab horizontal, the self-levelling cab operating limits, the speed of intervention, any effects on the levels of vibration. The ploughing and subsoiling operations were found to exhibit vibration exposure at low frequencies in the vicinity of natural frequencies of the human body and may consequently affect a driver's health and comfort [26].

## 2. Materials and Methods

### 2.1. The Tractor Used in the Tests

The tests were conducted on a tractor of medium–high power (147 kW) equipped with an automated self-levelling cab and with a high level of comfort for the driver. To provide the updated characteristic curves of the tractor's engine, aimed at quantifying the power available, the tractor was connected at a dynamometric brake (Borghi and Saveri, Bologna, Italy) [27]. Considering the subject of the tests, beyond the driver seat with pneumatic suspension (in which a rubber cylinder containing pressurized air works as a spring), the tractor had, as its most peculiar characteristic, an original hydraulic system operating the self-levelling of the cab, powered by the oil of the hydraulic system of the tractor (Figure 1).

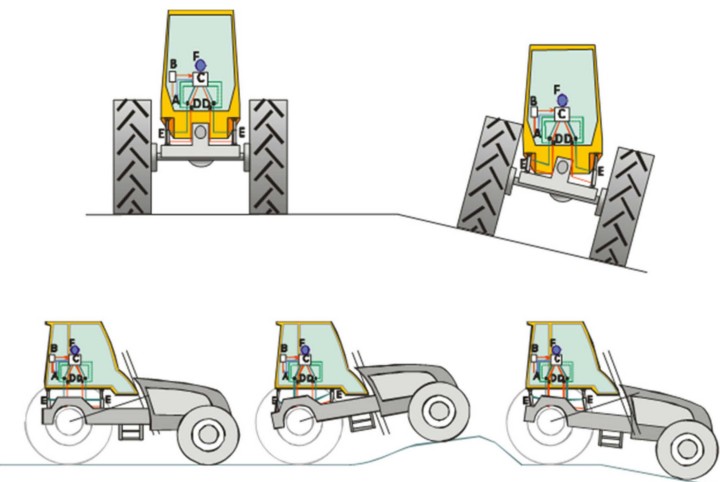

**Figure 1.** Sketch of the self-levelling cab according to the transversal and longitudinal slopes: (A) Oil flow from the reservoir and back; (B) hydraulic pump; (C) hydraulic distributor; (D) solenoid valves; (E) double effect cylinders; (F) gyroscopic sensor and computer.

The self-levelling apparatus is based on the presence of four hydraulic cylinders at the four corners of the cab floor, to form a square of 0.90 m side. A gyroscope detects changes in the slope of the ground according to the longitudinal and transverse directions and controls the action of the cylinders in such a way that the cab always remains horizontal. The maximum range of the cylinders, measured with the tractor stationary, is 0.23 m, corresponding to a maximum gradient of 25.5% and a maximum angle of 14.3° with respect to the two axes of the horizontal plane. The four cylinders are independent of one another and represent the only connection between cab and frame. The engine also has no point of contact with the interior. A silent block is mounted on each cylinder, with the function of reducing the vibrations, especially the high frequency ones (Figure 2). Therefore, the cabin, during the work, always remained parallel to a reference plane that forms a zero angle with the horizontal.

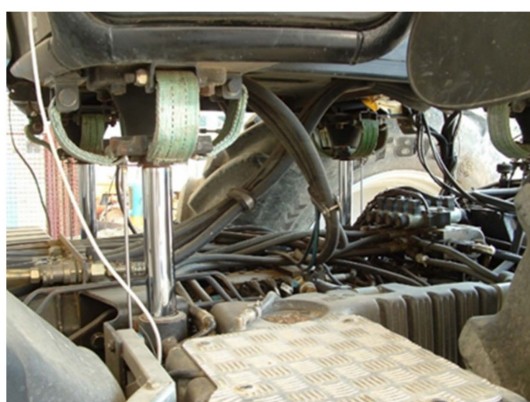

**Figure 2.** Particulars of a hydraulic cylinder and of the silent block on its top.

### 2.2. Instruments

The instrumental chain was composed of:

- Two six-channel signal conditioners Brüel & and Kjær;
- Eight-channel digital recorder;
- Signal acquisition and processing system Brüel & Kjær 5/1-ch. Input/Output Controller Module 0 Hz to 25.6 kHz frequency range (Figure 3a). The used sampling frequency was 160 Hz—suitable for analysing the level of vibration on tractors during field operations;

-  Tri-axial accelerometer adapted for driver seat Brüel & Kjær, type 4322 (Figure 3b) with relative calibrator, type 4294;
-  Two tri-axial accelerometers Brüel & Kjær, type 4321, positioned on the tractor chassis and on the cab floor (Figure 3c).

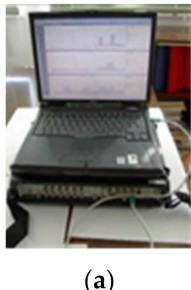
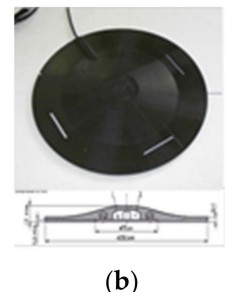
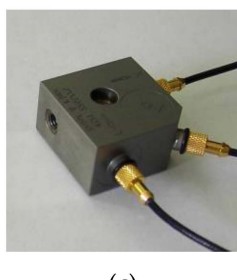

(**a**)                                                      (**b**)                                                      (**c**)

**Figure 3.** (**a**) Signal acquisition and processing system for data; (**b**) Three-axial accelerometer for driver seat; (**c**) Three-axial accelerometer placed on the cab floor.

*2.3. Measured Parameters, Data Processing and Reference Parameters*

The tests aimed to verify the presence of any effects due to the cab self-levelling system on the level of vibration transmitted to the whole-body of the tractor driver. The basic parameter to measure is the acceleration, a, expressed in m s$^{-2}$. As the effects of the vibrations depend on the frequency of the accelerations, these must be weighted by means of suitable filters according to the standard ISO 2631-1:1997, in order to quantify the WBV exposure in the reference frequency band 0.5–80 Hz, which is more dangerous for the human body in a sitting position. To observe how the vibrations are transmitted from the soil to the driver's seat through the various tractor elements, a tri-axial accelerometer was fixed on the driver's seat and the remaining two were placed, respectively, on the cab floor and on the tractor main frame, taking care that all three lay on the same vertical line. Vibration must be detected on three axes, defined by a coordinate system referring to the human body and originating at the point of contact between the subject and the vibrating surface. The *x*-axis passes through the back and chest, the *y*-axis through the shoulders, the *z*-axis through the feet and head (buttocks and head in the case of a seated person). As regards vehicle drivers, the x, y and z axes coincide with the longitudinal, transverse and vertical axes of the vehicle respectively. By adopting the risk assessment criterion for the health of seated subjects, according to the standard, the driver's seat was taken as the measurement plane and the triaxial seat accelerometer was positioned there, as shown in Figure 4.

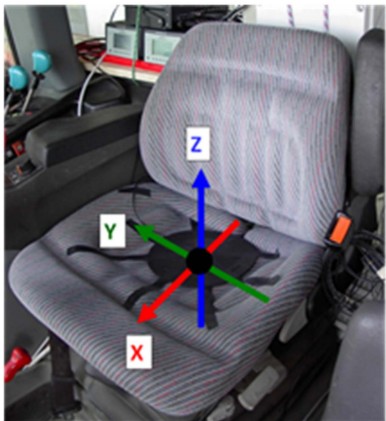

**Figure 4.** The driver seat equipped with tri-axial accelerometer and system of coordinates for the orientation of the accelerometer for seated position.

As regards the health, the evaluation criterion defined by the ISO 2631-1 standard refers to the various pathologies produced by vibrations and in particular to alterations of the spine. The criterion considers subjects regularly exposed to vibrations and concerns individuals in a seated position. The effects on health depend on the dose of vibrations absorbed, therefore the same effects correspond to the same dose.

Beyond the evaluation of vibrations with reference to their effects on health, the cited standard also states how to evaluate them in terms of effects on comfort, referring in this case to the means of transport where it is influenced by multiple factors. The interference between vibrations and certain activities (such as reading, writing, and drinking) may sometimes represent a cause of discomfort and involves the aspect of perception.

The comfort can be assessed through specific measurement of the accelerations on the three axes, on the seat cushion, on the seat back and under the feet. The ISO 2631-1 standard reports a vibration assessment scale formulated by passengers on public transport. The values refer to the overall sum of the vibrations.

Since these criteria can hardly be extended to our test conditions, it was decided to focus the analysis of the potential effects of the self-levelling cabin on the reduction of vibration levels harmful to human health, postponing the in-depth analysis on the effects on comfort to a later stage.

Since the effects of vibrations vary depending on the frequency of the acceleration, this must be weighted in frequency. The sensitivity of the human body to vibrations is at a maximum within a certain frequency range and gradually decreases, moving away from its lower and upper limits. Therefore, the weighting works by letting the signal generated by the accelerometer (analysed in frequency) pass unaltered in the range of maximum sensitivity and in attenuating it to a more or less progressive extent externally. In our case, the $W_d$ filter was adopted for the $x$ and $y$ axes, while the $W_k$ filter was applied for the z axis. This processing provided, for each axis, the frequency weighted acceleration, $a_w$:

$$a_w = \left[ \frac{1}{T} \int_0^T a_w^2(t)dt \right]^{1/2,} \tag{1}$$

where: $a_w$ (t) is the measured value of the acceleration; T is the acquisition time interval (s). The components of the acceleration along the three axes are simultaneously measured. The resultant vector of the acceleration, $a_v$, is provided by the relation:

$$a_v = \left( k_x^2 \, a_{wx}^2 + k_y^2 a_{wy}^2 + k_z^2 a_{wz}^2 \right)^{1/2,} \tag{2}$$

where $a_{wx}$, $a_{wy}$, $a_{wz}$ are weighted RMS accelerations along the x, y and z axes; $k_x$, $k_y$, $k_z$ are indices the values of which were determined depending on the effects of the relative components of the acceleration on the health: for $k_x$ e $k_y$ a value of 1.4 is applied in the case of sitting positions, as they are equal to 1 for the upright position; $k_z$ is equal to 1 in both positions.

The exposure to the vibrations can be quantified by normalizing the value of the acceleration $a_v$, measured during the daily exposure time ($T_e$), referring to an 8-h time interval, according to the principle of "equal energy", providing the normalized acceleration, A(8) according to the formula:

$$A(8) = a_w max \sqrt{\frac{T_e}{8}}. \tag{3}$$

However, as to the whole-body vibrations, the calculation of A(8) aimed at the health risk assessment is normally made only by considering the higher axial $a_w$ component. The determination of A(8) and the calculation of the maximum daily exposure times were performed using an Excel datasheet.

### 2.4. Soil Tillage Tests

The tractor was used in soil tillage tests, choosing operations which, involving the inclination of conventional cabs such as in-furrow ploughing or working on slopes, emphasize the interventions of the self-levelling system. Therefore, the tractor was firstly employed in the execution of in-furrow ploughing at a depth of 0.35 m with a three mounted furrow plough, on both plain and hilly surfaces. Then, on the same surfaces of the ploughing tests, the tractor was used to operate a mounted subsoiling plough at a depth of 0.20 m (Figure 5), which represented a further severe test condition [28,29].

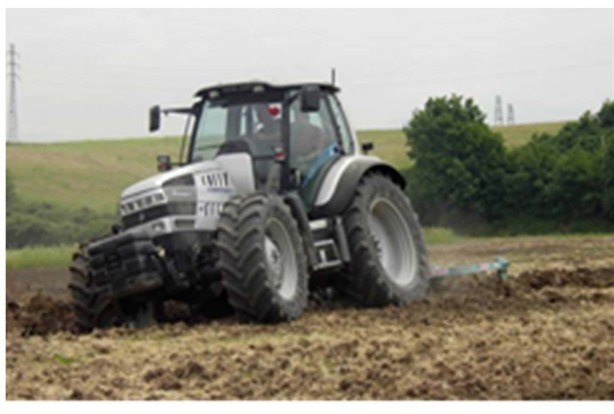
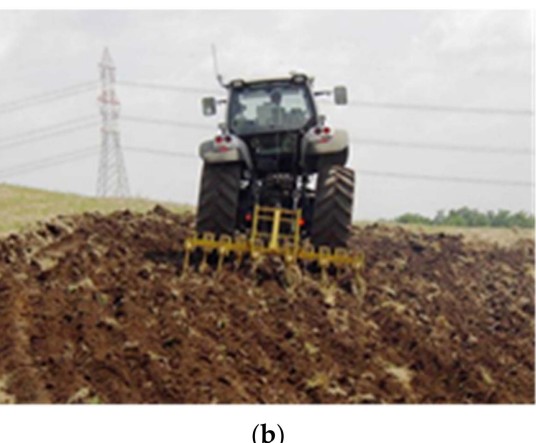

(**a**)                                                                                                   (**b**)

**Figure 5.** The tractor equipped with the self-levelling cab during the tests with the plough (**a**) and the subsoiling plough (**b**).

As regards the hill trials, a very demanding trajectory was chosen, with slopes (longitudinal and transversal) varying up to a maximum of over 45%. The operations were carried out transversely with reference to the dominant hillside slope, to highlight the operational capabilities of the self-levelling system.

As for the gear box ratios used in the tests, they were chosen to maximize, in each condition, the working speed. The sampling time intervals had the following mean durations: ploughing in plain: 94.2 s; Subsoiling in plain: 68.0 s; Ploughing on slope: 80.9 s; Subsoiling on slope: 80.7 s. The operations were executed with the self-levelling system disconnected (conventional mode, "OFF mode") and connected (system working, "ON mode") with the aim of comparing the results of the measurements of the levels of vibrations. Three replications were made for each test condition.

### 3. Results and Discussion

The results of the measurements of the levels of vibration during the operations described above are shown in Table 1 and in Figures 6 and 7.

In Table 1 it can be observed that the peaks of instant axial accelerations always occurred in the low frequency range, with small differences among chassis, cab floor and seat. It can be noticed that in the X and Y axes frequencies from 1 Hz up to 1.6 Hz were involved, without evident differences between OFF/ON, ploughing/subsoiling, in-plain/hillside. Higher peak frequencies were observed in the Z axis, with values frequently ranging from 2.5 Hz up to 6 Hz. Moreover, the ploughing peak frequencies were higher than those of subsoiling and the hillside tillage peak frequencies were higher than in-plain tillage frequencies.

As regards the trend of the weighted r.m.s. axial accelerations, the component $a_{wz}$ at the driver seat is always lower than $a_{wx}$ and $a_{wy}$ because of the damping action of the seat suspension on the Z axis.

**Table 1.** Tillage tests in plain and hillside. $a_{wx}$, $a_{wy}$, $a_{wz}$: weighted r.m.s. axial accelerations; Dom.: dominant component (in bold character), among $a_{wx}$, $a_{wy}$, $a_{wz}$, multiplied by 1.4; Res.: resultant vector of the acceleration, $a_v$; $T_e$: time of exposure as a function of "Dom." and "Res; $a_{peak}$: maximum observed instant acceleration; $F_{peak}$: frequency in correspondence of $a_{peak}$; ST: safety time (A(8) = 0.5 m s$^{-2}$); *LT*: limit time (A(8) = 1 m s$^{-2}$). The differences between ON and OFF modes are also reported for the accelerations (at chassis, cab floor and seat) and for ST and LT (only at driver seat).

| Tillage | Tillage Mode | Sensor's Position | X Axis $a_{wx}$ (m s$^{-2}$) | $a_{peak}$ (m s$^{-2}$) | $F_{peak}$ (Hz) | Y Axis $a_{wy}$ (m s$^{-2}$) | $a_{peak}$ (m s$^{-2}$) | $F_{peak}$ (Hz) | Z Axis $a_{wz}$ (m s$^{-2}$) | $a_{peak}$ (m s$^{-2}$) | $F_{peak}$ (Hz) | Dom. $1.4 \cdot a_{wmax}$ (m s$^{-2}$) | $T_{e(Dom.)}$ ST (h:min) | LT (h:min) | Res. $a_v$ (m s$^{-2}$) | $T_{e(Res.)}$ ST (h:min) | LT (h:min) |
|---|---|---|---|---|---|---|---|---|---|---|---|---|---|---|---|---|---|
| **Ploughing in plain** | Off | Seat | **0.38** | **0.22** | 1.2 | 0.36 | 0.2 | 1.2 | 0.28 | 0.14 | 2.5 | 0.53 | 6:31 | 28:16 | 0.78 | 2:60 | 12:60 |
| | | Cab floor | 0.3 | 0.16 | 1.2 | 0.29 | 0.17 | 1.2 | 0.29 | 0.13 | 2.5 | 0.42 | - | - | 0.65 | - | - |
| | | Chassis | 0.28 | 0.15 | 1.2 | 0.23 | 0.14 | 1.2 | 0.28 | 0.12 | 2.5 | 0.39 | - | - | 0.58 | - | - |
| | On | Seat | **0.37** | 0.2 | 1.4 | 0.33 | 0.17 | 1.0 | 0.27 | 0.15 | 2.5 | 0.52 | 6:52 | 29:49 | 0.74 | 3:19 | 14:25 |
| | | Cab floor | 0.26 | 0.13 | 1.4 | 0.24 | 0.14 | 1.0 | 0.3 | 0.13 | 2.5 | 0.30 | - | - | 0.58 | - | - |
| | | Chassis | 0.4 | 0.26 | 1.1 | 0.17 | 0.1 | 1.0 | 0.28 | 0.12 | 2.5 | 0.56 | - | - | 0.67 | - | - |
| | Diff. On-Off | Seat | −0.01 | - | - | −0.03 | - | - | −0.01 | - | - | −0.01 | 0:21 | 1:33 | −0.04 | 0:20 | 1:25 |
| | | Cab floor | −0.04 | - | - | −0.05 | - | - | 0.01 | - | - | −0.12 | - | - | −0.07 | - | - |
| | | Chassis | 0.12 | - | - | −0.06 | - | - | 0 | - | - | 0.17 | - | - | 0.09 | - | - |
| **Subsoiling in plain** | Off | Seat | **0.37** | 0.22 | 1.2 | 0.35 | 0.28 | 1.7 | 0.19 | 0.1 | 1.0 | 0.52 | 6:52 | 29:49 | 0.74 | 3:23 | 14:42 |
| | | Cab floor | 0.5 | 0.28 | 1.2 | 0.62 | 0.41 | 1.0 | 0.49 | 0.2 | 2.2 | 0.87 | - | - | 1.22 | - | - |
| | | Chassis | 0.41 | 0.24 | 1.2 | 0.57 | 0.34 | 1.7 | 0.32 | 0.16 | 1.0 | 0.80 | - | - | 1.03 | - | - |
| | On | Seat | **0.39** | 0.25 | 1.3 | 0.23 | 0.17 | 1.6 | 0.15 | 0.08 | 1.0 | 0.55 | 6:11 | 26:50 | 0.65 | 4:21 | 18:51 |
| | | Cab floor | 0.46 | 0.26 | 1.0 | 0.56 | 0.39 | 1.0 | 0.43 | 0.17 | 2.6 | 0.78 | - | - | 1.10 | - | - |
| | | Chassis | 0.35 | 0.19 | 1.0 | 0.49 | 0.28 | 1.6 | 0.28 | 0.13 | 1.0 | 0.69 | - | - | 0.89 | - | - |
| | Diff. On-Off | Seat | 0.02 | - | - | −0.12 | - | - | −0.04 | - | - | 0.03 | −0:41 | −2:59 | −0.09 | 0:58 | 4:10 |
| | | Cab floor | −0.04 | - | - | −0.06 | - | - | −0.06 | - | - | −0.08 | - | - | −0.12 | - | - |
| | | Chassis | −0.06 | - | - | −0.08 | - | - | −0.04 | - | - | −0.11 | - | - | −0.15 | - | - |
| **Hillside ploughing** | Off | Seat | **0.34** | 0.19 | 1.0 | 0.30 | 0.14 | 1.1 | 0.28 | 0.12 | 2.5 | 0.48 | 8:08 | 35:18 | 0.69 | 3:50 | 16:37 |
| | | Cab floor | 0.24 | 0.13 | 1.0 | 0.24 | 0.14 | 1.1 | 0.27 | 0.11 | 5.0 | 0.27 | - | - | 0.55 | - | - |
| | | Chassis | 0.21 | 0.11 | 1.0 | 0.21 | 0.14 | 1.0 | 0.27 | 0.11 | 5.0 | 0.27 | - | - | 0.50 | - | - |
| | On | Seat | 0.33 | 0.18 | 1.1 | **0.35** | 0.2 | 1.0 | 0.29 | 0.12 | 4.3 | 0.49 | 7:41 | 33:19 | 0.73 | 3:26 | 14:53 |
| | | Cab floor | 0.25 | 0.13 | 1.2 | 0.31 | 0.2 | 1.0 | 0.34 | 0.14 | 6.3 | 0.34 | - | - | 0.65 | - | - |
| | | Chassis | 0.21 | 0.1 | 1.2 | 0.22 | 0.16 | 1.0 | 0.3 | 0.13 | 5.0 | 0.30 | - | - | 0.52 | - | - |
| | Diff. On-Off | Seat | −0.01 | - | - | 0.05 | - | - | 0.01 | - | - | 0.01 | −0:27 | −1:59 | 0.04 | −0:24 | −1:44 |
| | | Cab floor | 0.01 | - | - | 0.07 | - | - | 0.07 | - | - | 0.07 | - | - | 0.11 | - | - |
| | | Chassis | 0 | - | - | 0.01 | - | - | 0.03 | - | - | 0.03 | - | - | 0.03 | - | - |
| **Hillside subsoiling** | Off | Seat | **0.34** | 0.21 | 1.1 | 0.3 | 0.18 | 1.0 | 0.22 | 0.09 | 2.9 | 0.48 | 8:08 | 35:18 | 0.67 | 4:05 | 17:43 |
| | | Cab floor | 0.22 | 0.14 | 1.0 | 0.24 | 0.16 | 1.0 | 0.19 | 0.08 | 5.0 | 0.34 | - | - | 0.49 | - | - |
| | | Chassis | 0.18 | 0.11 | 1.0 | 0.18 | 0.13 | 1.0 | 0.2 | 0.08 | 5.0 | 0.20 | - | - | 0.41 | - | - |
| | On | Seat | **0.36** | 0.23 | 1.1 | 0.33 | 0.22 | 1.0 | 0.23 | 0.1 | 2.3 | 0.50 | 7:15 | 31:30 | 0.72 | 3:33 | 15:22 |
| | | Cab floor | 0.22 | 0.15 | 1.1 | 0.26 | 0.19 | 1.0 | 0.2 | 0.08 | 4.0 | 0.36 | - | - | 0.52 | - | - |
| | | Chassis | 0.18 | 0.12 | 1.1 | 0.18 | 0.13 | 1.0 | 0.2 | 0.08 | 5.0 | 0.20 | - | - | 0.41 | - | - |
| | Diff. On-Off | Seat | 0.02 | - | - | 0.03 | - | - | 0.01 | - | - | 0.03 | −0:53 | −3:49 | 0.05 | −0:32 | −2:21 |
| | | Cab floor | 0 | - | - | 0.02 | - | - | 0.01 | - | - | 0.03 | - | - | 0.02 | - | - |
| | | Chassis | 0 | - | - | 0 | - | - | 0 | - | - | 0.00 | - | - | 0.00 | - | - |

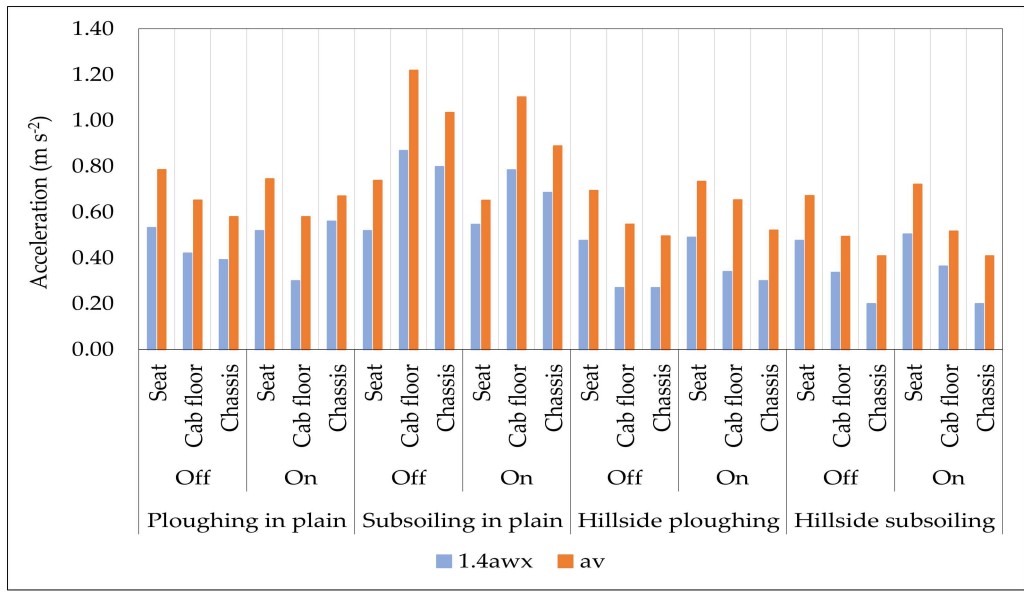

**Figure 6.** Average values of the vector acceleration, av, in the three points of application of the accelerometers in the different soil tillage.

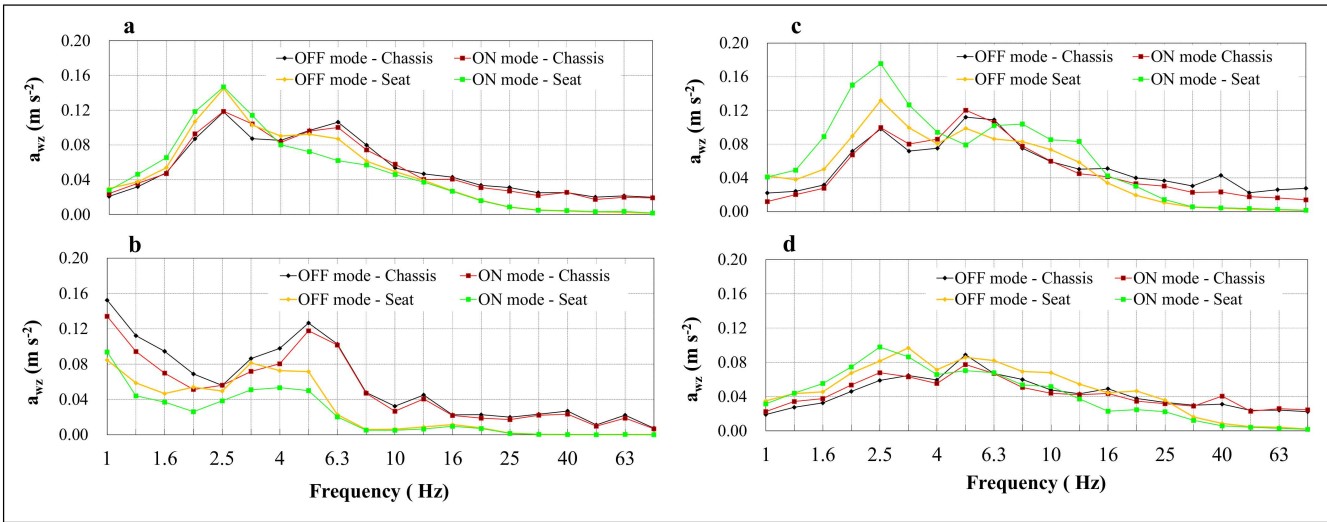

**Figure 7.** Frequency analysis of the acceleration along the Z axis during the four tillage tests, at the driver seat and at the chassis of the tractor. (**a**) ploughing in plain; (**b**) subsoiling in plain; (**c**) hillside ploughing; (**d**) hillside subsoiling.

Considering the ploughing in-plain, the levels of vibration at the driver seat are always lower in "ON mode" than in "OFF mode". In "ON mode", despite at frame level the components $a_w$ resulted in being higher on all axes; the corresponding $a_w$ values at the cab floor and at the seat resulted in being lower than in "OFF mode".

As to the hillside ploughing, despite similar $a_w$ at the chassis in both modes, as $a_{wx}$ decreased in ON mode, the combined effect of in-furrow ploughing, high soil cloddiness and cross slope caused an increase in $a_{wy}$ and $a_{wz}$ at cab floor and seat. This is particularly evident on the Y axis and is probably caused by the interventions of the self-levelling system to compensate to restore the cab horizontality, which could result in frequently not being adequate in terms of speed [30–32].

The vibration levels measured during the subsoiling in-plain show values of the components $a_w$ at the chassis in ON mode far lower than in OFF mode, which makes the

$a_w$ comparison between the two modes at cab floor and at seat level difficult. However, $a_{wx}$ was higher in ON mode than in OFF mode, while $a_{wy}$ and $a_{wz}$ were higher in OFF mode.

During the hillside subsoiling, despite similar acceleration values observed at the chassis in both modes, the solicitations at the seat in ON mode increased on all axes, determining the worsening of the working conditions.

Table 1 shows, in bold characters, the components of the accelerations which were dominant in each test condition. The component on the X axis was dominant in all tests except in hillside ploughing, where $a_{wy}$ prevailed. Probably, the variations in traction force during the tillage determined prevalent solicitations along the travel direction (the X axis), but in hillside ploughing in ON mode, they were overtaken by the transversal ones (Y axis) deriving from the interaction of transversal slope, in-furrow ploughing, great clods, and cabin movements operated by the self-levelling system. In fact, during this test, the slope sometimes exceeded the operational limit of the system (25.5% or 14.3°, see point 2.1) that was frequently disconnected and reconnected in an attempt to maintain the horizontality.

It can also be noticed, however, that in many cases the aforesaid dominants are not clearly higher than the other components whose values are often very close to those of the former. In particular, when $a_{wx}$ is dominant, $a_{wy}$ is the component with the closest value. Conversely, if $a_{wy}$ prevails, $a_{wx}$ follows it closely. This is important because of the higher relative weight of such components in WBV risk assessment according to the ISO 2631-1:1997. The values of $a_{wz}$ are always more distant, probably due to the effective action exerted by the seat suspension on the Z axis. However, with respect to the dominants, they do not appear to be negligible as well.

For these reasons, in addition to the safety time (ST) and limit time (LT) commonly calculated, according to the above standard, basing on the dominant components, Table 1 also shows the ST and LT values based on the vector sum, $a_v$, as stated by the same standard when the vibration in two or more axes is comparable [33], in order to assess how the three components affect the exposure time. The A(8) reference values of 0.5 m s$^{-2}$ and 1 m s$^{-2}$ were adopted respectively as safety value and limit value.

In Figure 6, the dominant axial component ($a_{wx}$ or $a_{wy}$) multiplied by the coefficient 1.4 is compared to the values of the resultant acceleration, $a_v$, calculated according to relation (2). It can be noticed that the latter is always significantly higher than the corresponding values of the former and, apart from these differences, that their trends are not always coherent in the different tests. For instance, in "subsoiling in plain", passing from OFF mode to ON mode, $a_{wx}$ increases, while $a_v$ decreases. This confirms that, with reference to the dominant, when non-dominant axial components are relatively high, they play an important role in determining the general level of vibration. Excluding them from the computation of the exposure time could lead to underestimation in WBV risk assessment and/or to different conclusions in the evaluation of an equipment such as the subject of this study. Said considerations clearly reflect on the values of ST and LT, which will result longer or shorter (Table 1) if they are calculated, respectively, on 1.4·$a_{wmax}$ or, more prudentially, on $a_v$.

The diagrams of Figure 7 show the frequency analyses of the accelerations on the Z axis measured at the driver seat and at the chassis (which reflects the soil unevenness) during the four tillage tests.

The curves in OFF and ON mode at the chassis have similar trends in each diagram, as some differences can be observed at seat level and can be attributed to the action of the self-levelling system combined with the seat suspension. Another general indication is that the frequency of the peak acceleration at the seat is lower than the frequency of the peak acceleration at the chassis. In both ploughing tests, the peaks at the seat are higher than at the chassis [34]. Then, in particular:

- Ploughing in-plain (Figure 7a): the curves at the driver seat in OFF and ON modes are very similar with their peak at 2.5 Hz, with values higher than those at the chassis. Small differences can be noticed for frequencies lower than 4.0 Hz, where

the accelerations for "ON mode" are slightly greater, and in the interval 4.0–12.5 Hz, where the values of "ON mode" are lower;

- Hillside ploughing (Figure 7c): the differences between OFF and ON have widened: "ON mode" clearly shows higher acceleration than OFF mode in the interval 1.0–4.0 Hz and 6.3–16.0 Hz. Additionally in this case, the peaks occurred at 2.5 Hz;

- Subsoiling in-plain (Figure 7b): even if the curves of the Z-acceleration at the chassis have similar shapes (with peaks at 5.0 Hz), below 6.3 Hz they have lower values in ON mode than in OFF. This is probably due to differences in soil unevenness and is reflected by the curves at the seat where, in the interval 1.0–6.3 Hz, the ON mode acceleration is much less than in "OFF mode". The shapes of the curves of the acceleration at the seat are different from those at the chassis, with peaks at 1.0 Hz in both theses;

- Hillside subsoiling (Figure 7d): in this case, the curves of the acceleration at the chassis have similar shapes (with peaks at 5.0 Hz), with small differences in the interval 1.0–6.3 Hz where the values of the "ON mode" are slightly higher than in A. At seat level, in the interval 1.0–5.0 Hz, the accelerations are higher than at the chassis and the "ON mode" shows worse behaviour (with peak at 2.5 Hz) than the "OFF mode" (with peak at 3.15 Hz).

## 4. Conclusions

The results of first test on a system for the self-levelling of an agricultural tractor cab indicate that it could contribute to improve the health preservation and comfort of the driver and, ultimately, to enhancing working conditions. The system has the function of maintaining the spine in a vertical position regardless of the slope of the ground, limiting damages resulting from stresses caused by uncomfortable posture (e.g., curvature of the lumbar region). The test highlighted both some operational situations in which the system improved the driver working conditions, and other situations that have been found to be critical for it and require further studies.

As to the mechanical performance, the self-levelling system proved to work well inside its operating limits of slope, operating gradually to keep the cab horizontal.

From the point of view of the axial accelerations, the horizontal components (X and Y) always resulted in being higher than the vertical one which was never dominant. The activation of the self-levelling system did not show univocal supportive effects, sometimes causing an increase in dominant components, mainly in hillside tests where the presence of high and variable slopes, often exceeding the system's limit, caused its frequent automatic switching on and off. This, combined with soil unevenness, particularly during the in-furrow ploughing, reduced the level of comfort (as the measured values of the accelerations demonstrated). As a consequence, the driver disconnected the system and continued to work in the traditional way. The use of the self-levelling system could be useful in hilly conditions, when the work occurs according to the level lines, or straight down, when the slope is quite constant. An additional, advisable modification should concern the sensitivity of intervention, which should not take into account the punctual soil unevenness (clods) but only the surface general behaviour (slope).

Conversely, during tilling in-plain, in conjunction with the system activation, a certain reduction of the vibration levels was observed, more evident in terms of the resultant acceleration $a_v$, compared to the traditional condition.

In general, the results show that the tillage is an activity characterized by significant solicitations (vibration and jolts) occurring on all spatial axes: beyond the dominant axial acceleration, the other components often have comparable values. Ignoring them could lead to an important underestimation of $T_e$. Therefore, the calculation of the exposure time to vibrations during agricultural work should be made based on the vector sum, $a_v$, which always resulted in being much greater than the axial accelerations $a_{wmax}$, even multiplied by the factor k = 1.4. This would provide a more prudent health risk assessment and a more correct evaluation of the performance of instrumentations and equipment such as

the subject of this study. As an example, Table 1 shows that the $T_e$ values based on $a_v$ are, meanly, about 50% of those obtained with $a_{wmax}$.

The last consideration is about spine posture during the work. If the self-levelling system is working correctly, the Z axis at the seat (perpendicular to the sitting plane) and the actual vertical line coincide. Under conventional working conditions, when the cab is not horizontal, the Z axis at the seat forms a solid angle with the actual vertical line (for instance, a transversal angle of 9.21° during in-furrow ploughing tests in-plain). The driver tends to compensate for this angle by arching the spine, causing unsuitable load distribution on the inter-vertebral disks. In such a situation, it is possible that the X and Y components also contribute to the actual vertical result of vibration. In this case, in order to avoid an underestimation of the risk from exposure to vibration, the X and Y components should contribute to the calculation of the time of daily exposure ($T_e$). All three axial components $a_w$ are also used in the assessment of the driver's comfort level, which will be the subject of a dedicated study. However, an exhaustive evaluation of these aspects will be possible through a multi-disciplinary approach involving the sectors of agricultural engineering (mechanics) and medicine (occupational medicine, orthopaedics).

**Author Contributions:** Conceptualization, D.P. and R.F.; methodology, L.F.; software, R.G.; validation, D.P., R.F. and L.F.; formal analysis, L.F.; data curation, L.F. and G.V.; writing—original draft preparation, R.F. and D.P.; writing—review and editing, R.F.; supervision, D.P.; funding acquisition, D.P. All authors have read and agreed to the published version of the manuscript.

**Funding:** This paper was funded with the contribution of the Italian Ministry of Agricultural, Food, Forestry and Tourism Policies (MiPAAFT) sub-project "Tecnologie digitali integrate per il rafforzamento sostenibile di produzioni e trasformazioni agroalimentari (AgroFiliere)" (AgriDigit program) (DM 36503.7305.2018 of 20/12/2018).

**Institutional Review Board Statement:** Not applicable.

**Informed Consent Statement:** Not applicable.

**Data Availability Statement:** Available data are contained within the article.

**Acknowledgments:** We thank Cesare Cervellini and Gino Brannetti (CREA-IT) for their precious technical support during the realization of the tillage test.

**Conflicts of Interest:** The authors declare no conflict of interest.

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
