# Peer review of "Levels of Whole-Body Vibrations Transmitted to the Driver of a Tractor Equipped with Self-Levelling Cab during Soil Primary Tillage"

_agriengineering, doi:10.3390/agriengineering4030044_

Round 1

Reviewer 1 Report

The work presented for review, entitled "Levels of whole-body vibrations transmitted to the driver of a tractor equipped with self-levelling cab during soil primary tillage" presents the results of tests on a tractor equipped with a self-levelling cab in order to determine its operational limitations, the best performance conditions for keeping the cab level, the intervention speed, the possible influence on transmitted whole-body vibration (WBV) levels, during initial soil tillage with a blade plough and a deep plough on both flat and hilly surfaces.

In my opinion, the work can be published as presented. 

Analysing the individual chapters in detail, I can say that: 

1) The introduction explains the topic and introduces the research issues that the authors examine in the publication. The Authors cite 27 literature references in the introduction. 

2) The chapter "Material and Methods" contains four subchapters. In the first "The tractor used in the tests" the Authors characterise the tractor used for the research. In the second, the Authors describe the instrumentation used in the study. In the third subchapter, the method of data acquisition is described and in the fourth subchapter, the Authors described the methodology for conducting the field study.

The methodology in my opinion is described correctly.

3) In the chapter Results and Discussion chapter, the Authors present the results in tables and figures.  Tables and figures are easy to read. The way the results are presented is not in doubt. 

It raises my doubts and I would suggest expanding the discussion of the results. The authors cited only one publication (31).

4) Conclusions based on research and not in doubt. 

5) Literature references collated as required by the publisher. 

The manuscript is also well edited, and I found no serious errors. 

I recommend the manuscript for publication. 

Reviewer 2 Report

Authors have measured the vibrations inside of a kind of cabin of a tractor. They have used ISO 2631 for studying vibrations from whole-body vibration point of view. It is an interesting topic.

The state of the art should include also references about other works based on UNE 2631 for others means of transportation (comfort studies), such as railways, cars, bicycles or e-scooters (10.13189/ujme.2022.100101, 10.25367/cdatp.2020.1.p141-147, 10.1007/s42417-021-00280-3 ….). Also, the measuring systems for measuring comfort (data acquisition systems to monitor comfort in cars, railways….).  Please, revise the norm ISO 2631-1:1997, there are more recent versions. Check it in all the paper and update it. Also, try to describe in more detail the given references. Avoid big groups of references without describe some of them (line 51, [11-16]).

In methods, it should be described the filter used for seated position, you could add a bode diagram and the main characteristics of this filter (from UNE 2631). When authors describe the weighted acceleration, the mathematics of UNE2631 could also add a reference to other papers where it has been already described (methodologies for the study of the influence vibrations on human comfort).

In methods, authors should explain in more detail the Directive 2002/44/EC.

Extent the Instruments section. You should add the used SAMPLING FREQUENCY.

There are no considerations about comfort classifications of the standard UNE-ISO 2631, please, check it and add also this kind of feedback. Also, I recommend to authors to revise papers what describe methodologies for the study of the influence vibrations on human comfort and to add an insight from the health point of view. Despite the very good systems for controlling the cabin, the accelerations are not very low, and the drivers of tractors use to work long journeys.

Improve the quality of Figure 1.

Round 2

Reviewer 2 Report

I find interesting for readers to add other means of transport in which ISO 2631 has been already used, although I agree you have presented State of the art related with agricultural machinery. My recommendation in this case is not mandatory, but I really believe your paper will improve with it.

The standard version can be used ISO 2631-1:1997, is equivalent to last ones as authors argue. Thus, it is correct to maintain it.

However, authors don’t have considered these comments:

·       Extent the Instruments section. You should add the used SAMPLING FREQUENCY. Very important point, not only to give the range, authors should give the exactly used sampling frequency.

·       There are no considerations about comfort classifications of the standard UNE-ISO 2631, please, check it and add also this kind of feedback. Also, I recommend to authors to revise papers what describe methodologies for the study of the influence vibrations on human comfort and to add an insight from the health point of view. Despite the very good systems for controlling the cabin, the accelerations are not very low, and the drivers of tractors use to work long journeys.

Round 3

Reviewer 2 Report

Authors have improve the paper and now, after the following corrcections, can be published:

.- As indicated in line 80, the standard considers frequencies up to 80Hz, thus, the used sample frequency should be al least 160Hz, even 200Hz, due the frequencies used in filters of the standard. Authors should justify this point (eg, based on the simulations, or other studies). If not, it could be a limitation to add in conclusions section.

.- Please, add the comfort study as possible future works in conclusion section.
